# The risk factors for treatment-related mortality within first three months after kidney transplantation

Ye Na Kim[1]☯, Do Hyoung Kim[2]☯, Ho Sik Shin[1], Sangjin Lee[3], Nuri Lee[4], Min-Jeong Park[4], Wonkeun Song[4], Seri Jeong[4]*

1 Division of Nephrology/Transplantation, Department of Internal Medicine, Kosin University College of Medicine, Gospel Hospital, Busan, South Korea, 2 Department of Internal Medicine, Hallym Kidney Research Institute, Hallym University College of Medicine, Seoul, South Korea, 3 Graduate School, Department of Statistics, Pusan National University, Busan, South Korea, 4 Department of Laboratory Medicine, Kangnam Sacred Heart Hospital, Hallym University College of Medicine, Seoul, South Korea

☯ These authors contributed equally to this work.
* hehebox@naver.com

**Data Availability Statement:** The data involved in this study have been deposited in Harvard Dataverse, and are accessible through https://doi.

## Abstract

Mortality at an early stage after kidney transplantation is a disastrous event. Treatment-related mortality (TRM) within 1 or 3 months after kidney transplantation has been rarely reported. We designed a cohort study using the national Korean Network for Organ Sharing database that includes information about kidney recipients between 2002 and 2016. Their demographic, and laboratory data were collected to analyze risk factors of TRM. A total of 19,815 patients who underwent kidney transplantation in any of 40 medical centers were included. The mortality rates 1 month (early TRM) and 3 months (TRM) after transplantation were 1.7% (n = 330) and 4.1% (n = 803), respectively. Based on a multivariate analysis, older age (hazard ratio [HR] = 1.044), deceased donor (HR = 2.210), re-transplantation (HR = 1.675), ABO incompatibility (HR = 1.811), higher glucose (HR = 1.002), and lower albumin (HR = 0.678) were the risk factors for early TRM. Older age (HR = 1.014), deceased donor (HR = 1.642), and hyperglycemia (HR = 1.003) were the common independent risk factors for TRM. In contrast, higher serum glutamic oxaloacetic transaminase (HR = 1.010) was associated with TRM only. The identified risk factors should be considered in patient counselling, and management to prevent TRM. The recipients assigned as the high-risk group require intensive management including glycemic control at the initial stage after transplant.

## Introduction

Patients, who underwent kidney transplantation, have better survival, improved cognition, and less economic burden than those who continue with dialysis [1–3]. Although kidney transplantation has improved during the past few decades [4], some kidney recipients still encounter early death after surgery, which is catastrophic for both the recipient and the medical staff.

org/10.7910/DVN/G4OLYV. We had accessed the Korean Network for Organ Sharing database from June 5, 2020 to September 23, 2020. Contact information for a data access committee is listed as follows: National Organ and Blood Management Institute of the Ministry of Health and Welfare, Tel: 82-2-2628-3602; Official internet site: https://www.konos.go.kr/konosis.

**Funding:** This work was supported by the National Research Foundation of Korea (NRF) grant, funded by the Ministry of Science, ICT & Future Planning (grant No. NRF-2017R1C1B2004597]. SJ received this fund. URL: http://www.nrf.re.kr/index The funders had no role in study design, data collection and analysis, decision to publish, or preparation of the manuscript.

**Competing interests:** The authors have declared that no competing interests exist.

**Abbreviations:** BUN, blood urea nitrogen; CI, confidence interval; Cl, chloride; HR, hazard ratio; K, potassium; KCDC, Korean Center for Disease Control; KONOS, Korean Network for Organ Sharing; Na, sodium; SGOT, serum glutamic oxaloacetic transaminase; SGPT, serum glutamic pyruvic transaminase; TRM, treatment-related mortality; WBC, white blood cell.

Treatment-related mortality (TRM), which is a different concept from disease-related mortality, is important value to improve survival after treatment. They provide information about factors that require intensive care and medical decisions during a critical period [5]. In major abdominal surgery, or cardiovascular procedures, 30-day mortality after surgery is defined as TRM [6–8]. In addition, 90-day postoperative mortality is a legitimate measure of TRM in hepatobiliary–pancreatic surgery [9]. Furthermore, the 90-day mortality rate is a good postoperative index predictor in colectomy, hepatectomy, and pneumonectomy [9–12]. Most studies have reported the results of kidney transplantation after 1 [13], 5 [14], or more than 10 years [15] previously. However, studies about 1- and 3-month mortality are seldom reported. Recently, the predictors related to TRM using the Health Insurance Review and Assessment Service were identified [16]. However, the lack of laboratory data limit the analysis of wider factors for TRM.

This study used a comprehensive database operated by the Korean Center for Disease Control (KCDC) that contains the medical records of all kidney recipients registered in the Korean Network for Organ Sharing (KONOS) system. Therefore, this database was suitable for our investigation of TRM.

Using this database, we conducted a comprehensive population-based analysis to investigate the risk factors of TRM after kidney transplantation. Our results would facilitate pre- and post-transplantation assessment and management, thereby contributing to improved outcome for kidney recipients.

## Materials and methods

### Study design

This was a retrospective and observational cohort study that used a prospectively registered national data set on transplantation. All patients who underwent kidney transplantation in 40 medical centers around the country between January 2002 and December 2016 were included. We defined death within 1 and 3 months after kidney transplantation as early TRM and TRM, respectively, and then investigated the risk factors for early TRM and TRM.

### Ethics statement

This study was performed in accordance with the Declaration of Helsinki and Istanbul, and approved by the independent Institutional Review Board of Kosin University Gospel Hospital (KUGH 2017-12-009). The need for informed consent was waived because anonymity of personal information was maintained.

### Study population

This study included all patients enrolled for kidney transplantation in the KONOS system of the KCDC between January 2002 and December 2016. We excluded patients who did not have complete demographic information and who concurrently underwent other organ transplantations [16]. During this period, 19,815 patients were enrolled in the database. A one-year washout period was applied to our data. All recipients were monitored from the time of registration for transplantation until death or until the study end date of December 2016. To manage the privacy risks, the database is managed by an authorized executive supervisor. We were allowed to perform this study through a research agreement with KONOS. The raw data were provided after de-identification. All analyses were performed without using any identifying process for recipients' personal information.

## Study variables

We collected the following demographic and clinical data about kidney recipients from the KONOS database: age; sex; donor status; weight; date of transplantation; any prior kidney transplant experience; and ABO compatibility. We also collected the following routine chemistry laboratory results: blood urea nitrogen, creatinine, glucose, albumin, protein, serum glutamic oxaloacetic transaminase (SGOT), serum glutamic pyruvic transaminase, total bilirubin. Electrolyte profile, hematology (white blood cell, hemoglobin, hematocrit, platelet) were also included. In this study, we included maximally available variables, which can be obtained from the KONOS database.

## Statistical analysis

Descriptive statistics are used for patient characteristics and clinical variables correlated with early TRM and TRM. Nominal and continuous variables were compared between groups using the chi-square test and Mann-Whitney U test, respectively. The median and interquartile range are used for non-normally distributed variables. To prevent confounding factors, univariate and multivariate Cox proportional-hazards regression models were used to examine the variables that correlate independently with TRM. Two-tailed $P$ values less than 0.05 were considered statistically significant.

Statistical analyses were performed using R statistical software, version 3.6.1 (R Foundation for Statistical Computing, Vienna, Austria), PASW software, version 18.0 (SPSS Inc., Chicago, IL, USA), and Analyse-it Method Evaluation Edition software, version 2.26 (Analyse-it Software Ltd., Leeds, UK).

## Results

### Characteristics of patients

We included 19,815 patients who underwent kidney transplantation between 2002 and 2016 in our study cohort. The baseline characteristics of these patients are presented in Table 1. The median age of the patients was 46.0 years (1st to 3rd quartile range: 36.0–53.0 years). Our cohort consisted of 11,750 men and 8,065 women. Most patients received a kidney from a living donor (62.2%), followed by deceased (37.5%) and cardiac death (0.3%) donors. Most recipients (92.5%) had no previous transplantation experience. ABO identical transplantation predominated (75.9%) over ABO compatible (17.1%) and ABO incompatible (7.0%) transplantations.

### Treatment-related mortality

Among 19,815 recipients, 330 (1.7%) and 803 (4.1%) died within 1 and 3 months after kidney transplantation, respectively. The overall cumulative incidence of mortality is shown in Fig 1A. The characteristics of kidney recipients who died within 1 and 3 months were compared with those of living patients and are summarized in Table 1. As shown by this comparative analysis, both early TRM and TRM increased significantly as age increased. In particular, among patients older than 60 years, the rates of those who died within 1 month (20.1%) and 3 months (17.8%) after transplantation were about two times higher than those of younger patients (9.5% for early TRM and 9.3% for TRM). Among the clinical data, age ($P < 0.001$), weight ($P = 0.007$ for early TRM and $P = 0.017$ for TRM), donor status ($P < 0.001$), a previous transplantation experience ($P = 0.010$ for early TRM and $P = 0.002$ for TRM), and ABO compatibility ($P < 0.001$) differed significantly between TRM and non-TRM patients. Among the laboratory variables, the levels of glucose ($P = 0.004$ for early TRM and $P = 0.002$ for TRM)

**Table 1. Characteristics of kidney recipients with treatment-related mortality after transplantation.**

| Variable[a] | Early TRM | | | TRM | | |
|---|---|---|---|---|---|---|
| | Alive at 1-month | Death by 1-month | *P*-value[b] | Alive at 3-months | Death by 3-months | *P*-value[b] |
| Age, years | 45.0 (36.0–53.0) | 52.0 (44.0–58.0) | < 0.001 | 45.0 (35.0–53.0) | 51.0 (43.0–58.0) | < 0.001 |
| Sex | | | | | | |
| Male | 11,545 (98.3) | 205 (1.7) | 0.319 | 11,277 (96.0) | 473 (4.0) | 0.845 |
| Female | 7,940 (98.5) | 125 (1.5) | 0.319 | 7,735 (95.9) | 330 (4.1) | 0.845 |
| Weight, kg | 60.0 (53.0–69.0) | 63.0 (55.0–70.0) | 0.007 | 60.0 (53.0–69.0) | 62.0 (53.0–70.0) | 0.017 |
| Donor | | | | | | |
| Living | 12,172 (98.8) | 152 (1.2) | < 0.001 | 11,935 (96.8) | 389 (3.2) | < 0.001 |
| Deceased | 7,254 (97.6) | 176 (2.4) | | 7,018 (94.5) | 412 (5.5) | |
| Cardiac death | 59 (96.7) | 2 (3.3) | | 59 (96.7) | 2 (3.3) | |
| Number of previous transplantations | | | | | | |
| 0 | 16,298 (98.7) | 221 (1.3) | 0.010 | 15,986 (96.8) | 533 (3.2) | 0.002 |
| ≥1 | 1,308 (97.8) | 30 (2.2) | | 1,273 (95.1) | 65 (4.9) | |
| ABO compatibility | | | | | | |
| ABO identical | 1,4803 (98.4) | 239 (1.6) | < 0.001 | 14,451 (96.1) | 591 (3.9) | < 0.001 |
| ABO compatible | 3,331 (98.6) | 47 (1.4) | | 3,262 (96.6) | 116 (3.4) | |
| ABO incompatible | 1,351 (96.8) | 44 (3.2) | | 1,299 (93.1) | 96 (6.9) | |
| Chemistry | | | | | | |
| BUN (mmol/L) | 19.6 (13.2–26.1) | 19.4 (11.8–25.4) | 0.430 | 19.6 (13.2–26.1) | 18.9 (11.4–24.3) | 0.091 |
| Creatinine (μmol/L) | 579.5 (404.1–838.8) | 556.6 (411.8–724.4) | 0.544 | 579.5 (404.1–838.8) | 533.8 (404.1–693.9) | 0.692 |
| Glucose (mmol/L) | 4.9 (4.1–6.3) | 5.4 (4.1–9.0) | 0.004 | 4.9 (4.1–6.2) | 5.2 (4.1–8.5) | 0.002 |
| Albumin (g/L) | 39.0 (35.0–42.0) | 38.0 (34.0–41.0) | 0.098 | 39.0 (35.0–42.0) | 38.0 (34.0–42.0) | 0.289 |
| Protein (g/L) | 67.0 (61.0–72.0) | 66.0 (58.0–73.0) | 0.235 | 67.0 (61.0–72.0) | 66.0 (58.0–73.0) | 0.402 |
| SGOT (μkat/L) | 0.3 (0.2–0.4) | 0.3 (0.2–0.4) | 0.001 | 0.3 (0.2–0.4) | 0.3 (0.2–0.4) | 0.010 |
| SGPT (μkat/L) | 0.2 (0.1–0.3) | 0.2 (0.1–0.4) | 0.156 | 0.2 (0.1–0.3) | 0.2 (0.1–0.4) | 0.773 |
| Total bilirubin (μmol/L) | 5.1 (5.1–8.6) | 6.8 (5.1–8.6) | 0.878 | 5.1 (5.1–8.6) | 6.8 (5.1–8.6) | 0.841 |
| Electrolyte | | | | | | |
| Na (mmol/L) | 138.0 (135.0–141.0) | 138.0 (133.9–141.0) | 0.779 | 138.0 (135.0–141.0) | 138.0 (133.0–141.0) | 0.518 |
| K (mmol/L) | 4.8 (4.2–5.3) | 4.7 (4.2–5.5) | 0.565 | 4.8 (4.2–5.4) | 4.7 (4.1–5.4) | 0.557 |
| Cl (mmol/L) | 17.0 (12.6–97.0) | 17.0 (12.0–97.0) | 0.686 | 17.0 (13.0–97.0) | 16.0 (12.0–95.0) | 0.059 |
| Hematology | | | | | | |
| WBC (x10^9/L) | 7.7 (4.8–55.1) | 8.7 (5.5–58.1) | 0.021 | 7.7 (4.7–55.3) | 8.6 (5.1–58.1) | 0.238 |
| Hemoglobin (g/L) | 89.0 (62.0–111.0) | 93.0 (19.0–114.0) | 0.986 | 89.0 (62.0–111.0) | 93.0 (19.0–112.0) | 0.957 |
| Hematocrit (proportion of 1.0) | 0.3 (0.2–0.3) | 0.3 (0.3–0.3) | 0.115 | 0.3 (0.2–0.3) | 0.3 (0.2–0.3) | 0.199 |
| Platelet (x10^9/L) | 177.0 (124.0–235.0) | 164.0 (97.0–238.0) | 0.204 | 177.0 (124.0–235.0) | 169.0 (95.3–238.0) | 0.469 |

[a]Data are expressed as number (%) or median (interquartile range).

[b]*P*-value was calculated using the chi-square test or Mann–Whitney *U* test.

Abbreviations: TRM, treatment-related mortality; BUN, blood urea nitrogen; SGOT, serum glutamic oxaloacetic transaminase; SGPT, serum glutamic pyruvic transaminase; Na, sodium; K, potassium; Cl, chloride; WBC, white blood cell.

and SGOT (*P* = 0.001 for early TRM and *P* = 0.010 for TRM) were both significantly higher in the early TRM and TRM groups than in the non-TRM group.

## Risk factors for early TRM and TRM

The risk factors for early TRM and TRM are presented in Tables 2 and 3, respectively. Based on the Cox multivariate analysis, older age (hazard ratio [HR] = 1.044; *P* < 0.001), deceased

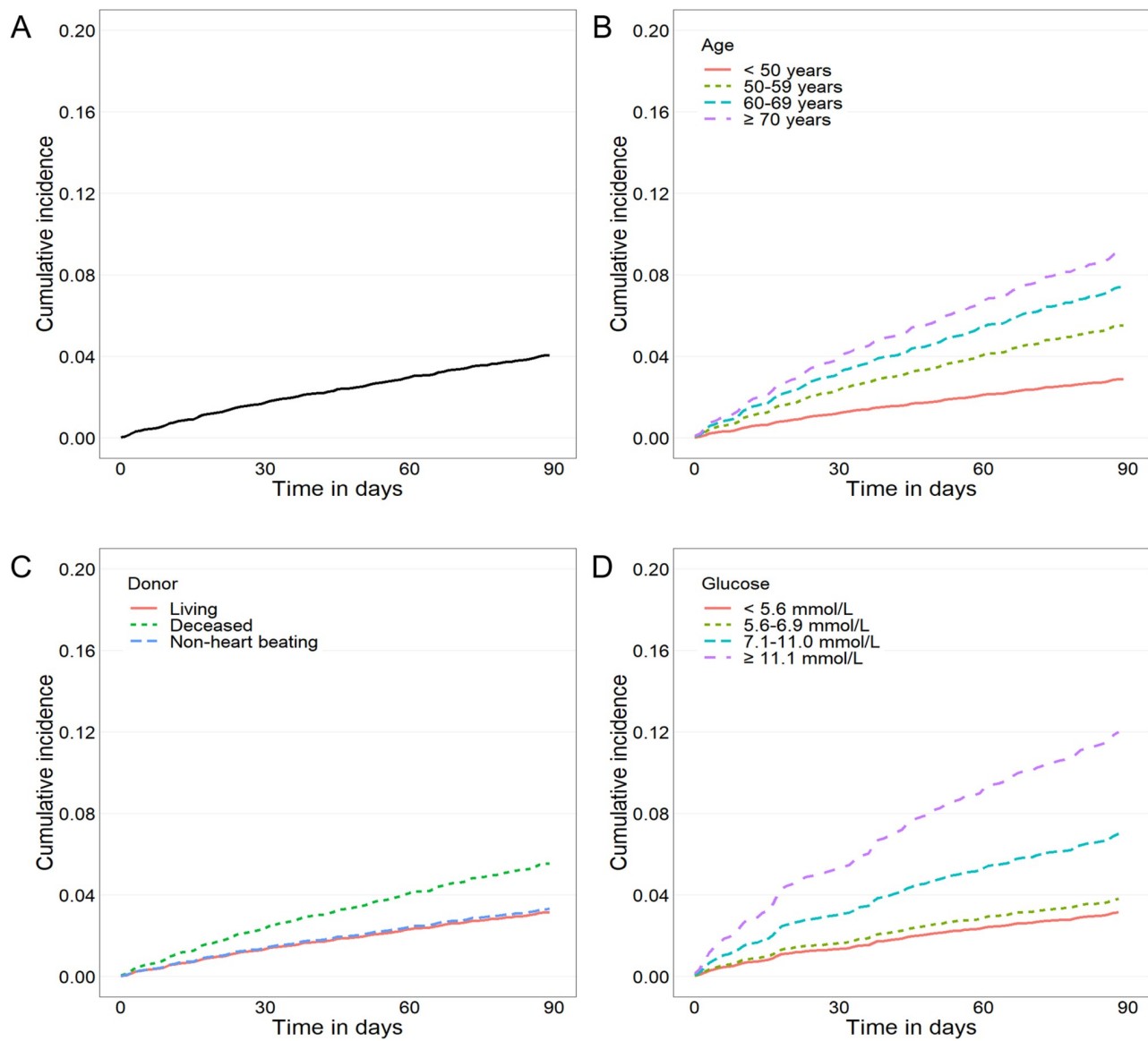

**Fig 1. Cumulative incidence of mortality according to independent factors common to both 1- and 3-month mortality after kidney transplantation.** (A) Total incidence. (B) Older age, (C) Deceased donor, and (D) Hyperglycemia were related to worse outcomes.

donor (HR = 2.210; $P < 0.001$), re-transplantation (HR = 1.675; $P = 0.007$), ABO incompatible transplantation (HR = 1.811; $P = 0.029$), higher glucose (HR = 1.002; $P = 0.047$), and hypoalbuminemia (HR = 0.678; $P = 0.046$) were independently associated with early TRM. Moreover, older age (HR = 1.014; $P = 0.010$), deceased donor (HR = 1.642; $P = 0.001$), and hyperglycemia (HR = 1.003; $P = 0.002$) were consistently independent risk factors for TRM at any time. Higher SGOT (HR = 1.010; $P = 0.009$) correlated only with TRM.

The effect of age on the cumulative incidence of mortality is presented in Fig 1B. The older age group presented with higher HRs for both early TRM (50–59 years, 2.293; 60–69 years, 3.254; and 70 years or older, 4.162; $P < 0.001$) and TRM (50–59 years, 1.947; 60–69 years, 2.632; and 70 years or older, 3.263; $P < 0.001$). The effects of donor status and glucose level on the cumulative incidences are shown in Fig 1C and 1D, respectively. In early TRM, a significant difference was observed between patients with and without previous transplantation

**Table 2. Univariate and multivariate analyses of 1-month mortality after kidney transplantation.**

| Variable | Univariate | | Multivariate[a] | |
|---|---|---|---|---|
| | HR (95% CI) | *P*-value | HR (95% CI) | *P*-value |
| Age, years | 1.045 (1.035–1.056) | < 0.001 | 1.044 (1.032–1.057) | < 0.001 |
| Sex | | | | |
| Male | Reference | | Reference | |
| Female | 0.887 (0.711–1.109) | 0.293 | 1.231 (0.920–1.646) | 0.162 |
| Weight, kg | 1.013 (1.004–1.021) | 0.004 | 1.011 (0.998–1.024) | 0.087 |
| Donor | | | | |
| Living | Reference | | Reference | |
| Deceased | 1.876 (1.517–2.320) | < 0.001 | 2.210 (1.625–3.005) | < 0.001 |
| Cardiac death | 2.541 (0.630–10.249) | 0.190 | 3.895 (0.952–15.946) | 0.059 |
| Number of previous transplantations | | | | |
| 0 | Reference | | Reference | |
| ≥1 | 1.684 (1.150–2.465) | 0.007 | 1.675 (1.151–2.438) | 0.007 |
| ABO compatibility | | | | |
| ABO identical | Reference | | Reference | |
| ABO compatible | 0.875 (0.640–1.196) | 0.402 | 1.127 (0.760–1.671) | 0.552 |
| ABO incompatible | 2.001 (1.451–2.760) | < 0.001 | 1.811 (1.062–3.086) | 0.029 |
| Chemistry | | | | |
| BUN | 0.995 (0.988–1.003) | 0.212 | | |
| Creatinine | 0.986 (0.942–1.033) | 0.563 | | |
| Glucose | 1.003 (1.001–1.005) | 0.002 | 1.002 (1.000–1.005) | 0.047 |
| Albumin | 0.676 (0.483–0.948) | 0.023 | 0.678 (0.462–0.994) | 0.046 |
| Protein | 0.893 (0.711–1.121) | 0.328 | | |
| SGOT | 1.007 (1.000–1.014) | 0.043 | 1.008 (0.994–1.022) | 0.245 |
| SGPT | 1.005 (0.995–1.015) | 0.323 | | |
| Total bilirubin | 0.719 (0.284–1.823) | 0.487 | | |
| Electrolyte | | | | |
| Na | 1.006 (0.999–1.012) | 0.076 | | |
| K | 0.981 (0.874–1.102) | 0.748 | | |
| Cl | 1.000 (0.995–1.006) | 0.941 | | |
| Hematology | | | | |
| WBC | 1.001 (1.000–1.001) | 0.056 | | |
| Hemoglobin | 0.962 (0.913–1.013) | 0.143 | | |
| Hematocrit | 1.005 (0.990–1.021) | 0.497 | | |
| Platelet | 1.000 (0.998–1.002) | 0.960 | | |

[a]Variables less than 0.05 of *P*-values in univariate analysis were included in the multivariate analysis.

Abbreviations: HR, hazard ratio; CI, confidence interval; BUN, blood urea nitrogen; SGOT, serum glutamic oxaloacetic transaminase; SGPT, Serum glutamic pyruvic transaminase; Na, sodium; K, potassium; Cl, chloride; WBC, white blood cell.

experience (Fig 2A). ABO incompatibility (Fig 2B) and serum albumin level (Fig 2C) also affected early TRM. The SGOT level correlated only with TRM, as shown in Fig 2D.

## Discussion

In this study, we conducted a comprehensive analysis of 1- and 3-month mortality after kidney transplantation in Korea. Older age, a deceased donor, and elevated glucose level were common risk factors for both early TRM and TRM. Re-transplantation, ABO incompatibility, and

**Table 3. Univariate and multivariate analyses of 3-month mortality after kidney transplantation.**

| Variable | Univariate | | Multivariate[a] | |
|---|---|---|---|---|
| | HR (95% CI) | *P*-value | HR (95% CI) | *P*-value |
| Age, years | 1.037 (1.031–1.044) | < 0.001 | 1.014 (1.003–1.025) | 0.010 |
| Sex | | | | |
| Male | Reference | | Reference | |
| Female | 1.017 (0.883–1.170) | 0.816 | 1.313 (0.994–1.735) | 0.055 |
| Weight, kg | 1.007 (1.002–1.013) | 0.011 | 1.007 (0.996–1.019) | 0.206 |
| Donor | | | | |
| Living | Reference | | Reference | |
| Deceased | 1.779 (1.549–2.043) | < 0.001 | 1.642 (1.211–2.226) | 0.001 |
| Cardiac death | 1.051 (0.262–4.216) | 0.944 | 1.852 (0.256–13.404) | 0.542 |
| Number of previous transplantations | | | | |
| 0 | Reference | | Reference | |
| ≥1 | 1.518 (1.174–1.964) | 0.001 | 1.121 (0.754–1.666) | 0.572 |
| ABO compatibility | | | | |
| ABO identical | Reference | | Reference | |
| ABO compatible | 0.872 (0.714–1.064) | 0.177 | 1.201 (0.810–1.779) | 0.362 |
| ABO incompatible | 1.778 (1.433–2.207) | < 0.001 | 1.514 (0.942–2.435) | 0.087 |
| Chemistry | | | | |
| BUN | 0.995 (0.991–1.000) | 0.044 | 0.994 (0.989–1.000) | 0.037 |
| Creatinine | 0.989 (0.961–1.019) | 0.469 | | |
| Glucose | 1.003 (1.002–1.005) | 0.000 | 1.003 (1.001–1.004) | 0.002 |
| Albumin | 0.830 (0.660–1.042) | 0.108 | | |
| Protein | 0.937 (0.817–1.074) | 0.351 | | |
| SGOT | 1.006 (1.001–1.011) | 0.021 | 1.010 (1.002–1.018) | 0.009 |
| SGPT | 1.003 (0.996–1.011) | 0.417 | | |
| Total bilirubin | 0.782 (0.453–1.350) | 0.377 | | |
| Electrolyte | | | | |
| Na | 1.003 (0.999–1.006) | 0.145 | | |
| K | 1.001 (0.984–1.018) | 0.898 | | |
| Cl | 0.998 (0.994–1.001) | 0.246 | | |
| Hematology | | | | |
| WBC | 1.000 (1.000–1.001) | 0.541 | | |
| Hemoglobin | 0.981 (0.950–1.014) | 0.260 | | |
| Hematocrit | 1.001 (0.989–1.014) | 0.833 | | |
| Platelet | 0.999 (0.998–1.001) | 0.340 | | |

[a]Variables with *P*-values less than 0.05 in the univariate analysis were included in the multivariate analysis.

Abbreviations: HR, hazard ratio; CI, confidence interval; BUN, blood urea nitrogen; SGOT, serum glutamic oxaloacetic transaminase; SGPT, Serum glutamic pyruvic transaminase; Na, sodium; K, potassium; Cl, chloride; WBC, white blood cell.

lower albumin correlated mainly with early TRM. In contrast, higher SGOT was associated with only TRM. According to recent systematic reviews, the incidence and mortality of acute kidney injury have an inverse correlation because of increased awareness and intensive management of acute conditions. Therefore, notification and care for the risk factors found in this study could contribute to improved outcomes.

Our risk analysis showed that age was a significant factor ($P < 0.001$) in both early TRM and TRM. The significant association between old age and poor outcome was persistently

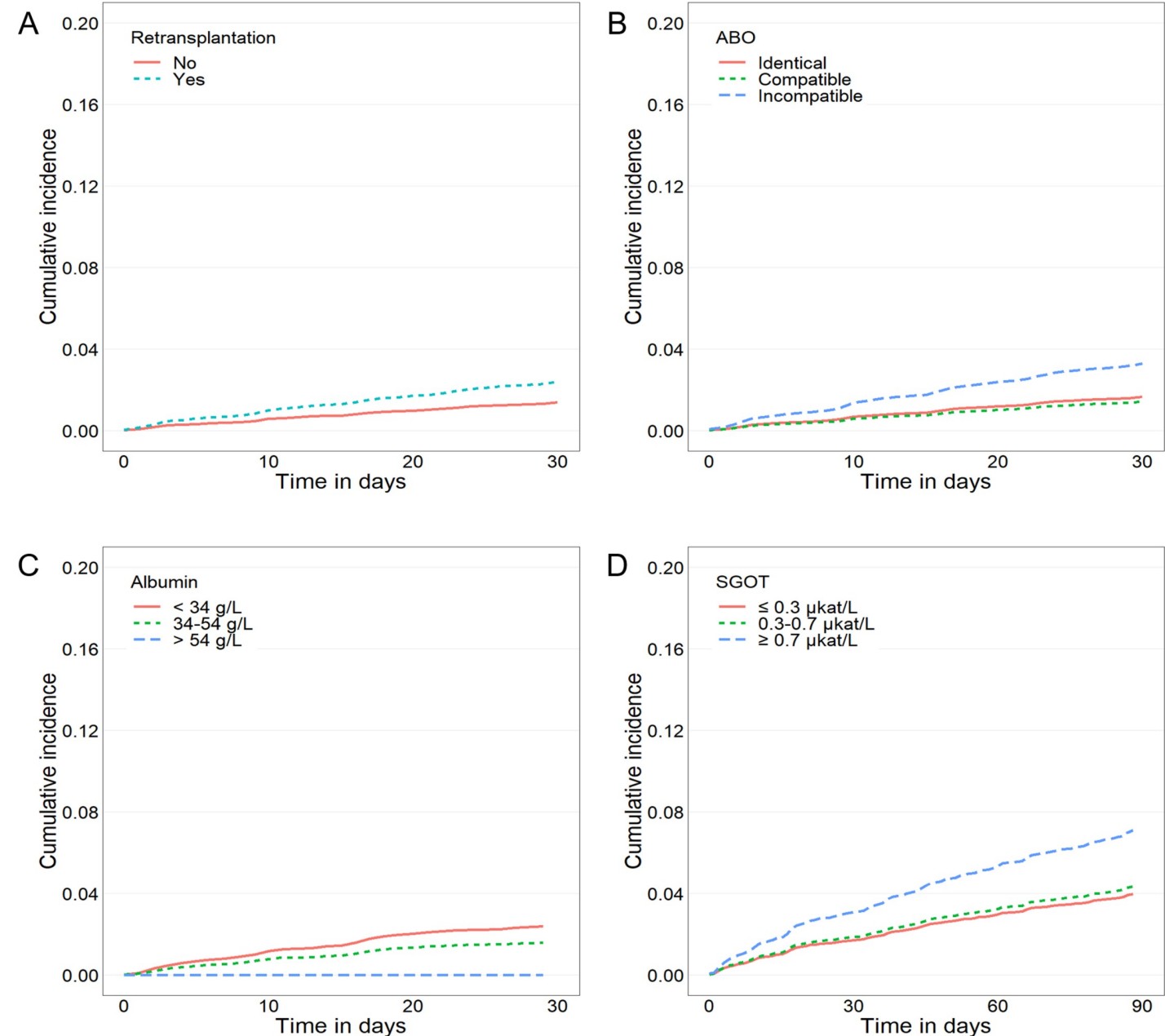

**Fig 2. Cumulative incidence of mortality according to the factors associated with 1- or 3-month mortality after kidney transplantation.** (A) Re-transplantation correlated with a worse outcome 1 month after transplantation. (B) ABO incompatibility and (C) Lowered albumin were risk factors for 1-month mortality after transplantation. (D) Higher serum glutamic oxaloacetic transaminase (SGOT) level was a predictor of 3-month mortality.

reported in previous studies [13, 17], and should thus be considered during patient counselling and selection.

Donor status has been a well-known, important factor in short- and long-term mortality after kidney transplantation [18]. According to previous studies, kidney allograft recipients, who died within the first year after transplantation were more likely to be recipients of deceased donor kidneys with longer duration of ESRD than live donor kidneys [13, 19]. It was difficult to compare TRM in our cohort with that in other countries because of a lack of

available data. When comparing 1-year allograft survival from a deceased donor, the survival of our cohort (82.9%) was worse than that reported in the United States (93.4%) and Europe (90.7%). More intensive care for recipients from deceased donors at an early point after transplantation is recommended. Recipients from cardiac death donors had higher incidences of graft loss and delayed graft function compared with recipients from living donors; however, the long-term kidney transplant outcomes with cardiac death donors and brain-dead donors were comparable in Western countries [20, 21]. The proportion of donation after cardiac death in our cohort (0.3%) was much smaller than in the United States (17.1%) [18] and the Netherlands (42.9%) [21]. Transplantation from cardiac death donors should be encouraged because those donors represent a potential solution to the imbalance between the number of end-stage kidney disease patients on waiting lists and the number of available kidney grafts.

Diabetes mellitus is a well-established risk factor for mortality after kidney transplantation [13, 22–24]. According to our results, prediabetic (5.6–6.9 mmol/L or 7.1–11.0 mmol/L of glucose) and diabetic (more than 11.1 mmol/L of glucose) [25] status was significantly correlated with TRM. A guideline development group in Europe recommends that diabetes in itself should not be considered a contraindication to kidney transplantation [26]. Because pre-emptive transplantation has a significant survival advantage over dialysis, patients with diabetes should be referred to transplant centers for early evaluation whenever feasible [27–29]. In addition, intensive management for glycemic control in patients with high glucose levels should be encouraged, particularly before and in the early stage after surgery because poor glycemic control after kidney transplantation is associated with poor outcomes [30].

Regarding re-transplantation, previous studies reported contradictory results. Re-transplantation was found to be a risk factor [22, 31] or a protective factor [32], and other studies showed that re-transplantation had no significant correlation with short- or long-term mortality [33, 34]. In our cohort, re-transplantation was a risk factor only for early TRM. The presence of immunologic risk factors such as prior sensitization, selection bias related to comorbidities, and the intensification of immunosuppression could cause these complex results [18, 35]. Taken together, we recommend re-transplantation only if intensive immunologic work-ups, monitoring, and management can be applied to recipients, especially in the first 30 days after transplantation.

The adoption of rituximab, plasmapheresis, and intravenous immunoglobulin enables ABO incompatible kidney transplantation [36]. According to a recent meta-analysis, recipients with ABO-incompatible kidney transplantation presented one-year graft survival (96%) slightly inferior to those who received an ABO-compatible transplant (98%). Most cases of mortality in ABO incompatible kidney transplantation occurred within 6 months [36], which is concordant with our results (related to early TRM only). The most common cause of death was infection, followed by antibody-mediated rejection, and bleeding [37]. Strong pre-transplant desensitization could be the cause of infection-related mortality during the early post-transplantation period. Therefore, reduced desensitization intensity and maintenance immunosuppression dose with concurrent immunologic monitoring, such as anti-A/B antibody titer, and patient-based blood transfusions are recommended for ABO incompatible recipients.

As a modifiable factor, control of hypoalbuminemia ($< 34$ g/L) [38] is important to prevent catastrophic early TRM. Hypoalbuminemia is frequently observed in hospitalized patients and can be related to several underlying diseases, including cirrhosis, poor nutritional status, inflammation, nephrotic syndrome, and sepsis. Regardless of its cause, low albumin levels on admission have a strong predictive value on short- and long-term mortality [38, 39]. According to a registry in the United States, every increase of 2 g/L in the pre-transplant serum albumin level was associated with a 13% decrease in all-cause mortality during follow up, a 17%

decrease in cardiovascular mortality, and a 4% decrease in delayed graft function risk [40]. The normalization of albumin levels, with care taken for underlying inflammation-related conditions such as improving the nutritional status of hemodialysis patients waiting for a transplant, is recommended to improve post-transplant outcomes.

The serum concentration of SGOT is routinely measured to assess liver function in pre-transplant patients. Aminotransferases are normally present in the circulation in low concentrations, usually < 40 U/L. However, the SGOT levels in patients with chronic kidney disease commonly decrease because of pyridoxine deficiency, a necessary coenzyme for SGOT, and the uremic environment [41]. A recent study revealed that increasing SGOT levels of > 20 U/L were incrementally and almost linearly associated with a higher death risk, and an increase of ≥ 40 U/L was associated with the highest risk of mortality (HR = 1.46) in hemodialysis patients [42]. Although reports investigating a direct association between SGOT and mortality are very rare, these findings in hemodialysis patients could be transferred to recipients of kidney transplantation. The assessment of liver function and timely improvement of liver disease could confer a survival benefit to kidney recipients.

This study had several limitations. Some variables in the KONOS database were missing data because entering all laboratory variables was not mandatory. That lack of information could have restricted our TRM analysis. Our results nonetheless offer preliminary evidence for selecting important variables that could be essential for the assignment of kidney transplants in the future. Moreover, we did not adjust for the causes of ESRD (confirmed by biopsy) or comorbidities in our TRM analyses because these data were not available from the KONOS database. Despite these limitations, the strengths of this study include the use of a nationwide population database of kidney recipients over a long time period. To the best of our knowledge, no other study has reported TRM risk factors using a nationwide data source, particularly in Asia. The relatively large sample size covering an entire national population and the unbiased measures used in this study thus provide reliable information about kidney recipients.

## Conclusions

In conclusion, our study characterized risk factors for 1- and 3-month mortality after kidney transplantation. Old age, particularly greater than 70 years, donor status, and a high glucose level prior to transplant were common risk factors for both early TRM and TRM. In contrast, re-transplantation, ABO incompatibility, and albumin concentration were risk factors for only early TRM, and a high serum SGOT level was an important risk factor for only TRM. Recipients with these risk factors require intensive management immediately after transplantation. To prevent catastrophic TRM, the factors we have identified should be considered when counselling and selecting patients for kidney transplants.

## Acknowledgments

The authors acknowledge the efforts of the staff of the KONOS database, which is supported by the KCDC, for the maintenance and extraction of precise data about kidney transplantation as a research resource. We also thank Hyun Jung Kim and Hyeong Sik Ahn and the staff of the Department of Preventive Medicine, College of Medicine, Korea University, for their assistance in preparing this article.

## Author Contributions

**Conceptualization:** Ye Na Kim, Ho Sik Shin, Seri Jeong.

**Data curation:** Sangjin Lee, Seri Jeong.

**Formal analysis:** Do Hyoung Kim, Sangjin Lee.

**Funding acquisition:** Seri Jeong.

**Investigation:** Ye Na Kim, Do Hyoung Kim.

**Methodology:** Do Hyoung Kim, Ho Sik Shin, Nuri Lee.

**Project administration:** Seri Jeong.

**Resources:** Do Hyoung Kim, Nuri Lee, Seri Jeong.

**Supervision:** Ye Na Kim, Ho Sik Shin, Seri Jeong.

**Validation:** Min-Jeong Park, Wonkeun Song.

**Visualization:** Do Hyoung Kim, Sangjin Lee, Seri Jeong.

**Writing – original draft:** Ye Na Kim, Do Hyoung Kim.

**Writing – review & editing:** Min-Jeong Park, Wonkeun Song, Seri Jeong.

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
