## [Decision Letter · Decision Letter 0]

27 Oct 2020

PONE-D-20-30070

The risk factors for treatment-related mortality in kidney transplantation using the Korean Network for Organ Sharing Database, 2002 to 2016

PLOS ONE

Dear Dr. Jeong,

Thank you for submitting your manuscript to PLOS ONE. After careful consideration, we feel that it has merit but does not fully meet PLOS ONE’s publication criteria as it currently stands. Therefore, we invite you to submit a revised version of the manuscript that addresses the points raised during the review process.

Please revise according to reviewers' suggestion.

We look forward to receiving your revised manuscript.

Kind regards,

Academic Editor

PLOS ONE

Journal Requirements:

2. Please include the date(s) on which you accessed the databases or records to obtain the data used in your study.

3. Please describe any inclusion and exclusion criteria used to determine your final cohort.

Reviewers' comments:

Reviewer's Responses to Questions

**Comments to the Author**

1. Is the manuscript technically sound, and do the data support the conclusions?

Reviewer #1: Yes

Reviewer #2: Yes

2. Has the statistical analysis been performed appropriately and rigorously? 

Reviewer #1: Yes

Reviewer #2: Yes

3. Have the authors made all data underlying the findings in their manuscript fully available?

Reviewer #1: Yes

Reviewer #2: Yes

4. Is the manuscript presented in an intelligible fashion and written in standard English?

Reviewer #1: Yes

Reviewer #2: Yes

5. Review Comments to the Author

Reviewer #1: Treatment-related mortality (TRM) after renal transplantation is a concept different from disease-related mortality and appeared to be a very prevalent entity.

Strengths:

- There are scarce data in scientific literature about TRM within 1 or 3 months after kidney transplantation.

- Authors collected an important amount of data from a very large cohort of patients using a national population based database, which included information about a total of 16,073 kidney recipients.

This article would be a valuable contribution to the medical literature to encourage further discussion on this entity.

The writing is clear and easily understandable.

The Authors have worked hard to improve this article, and they have met all criticisms raised by referees.

I feel the manuscript is now suitable for publication in Plos One.

Probably a better title could include the concept of an early mortality after renal transplant, and could exclude the origin of the database and the years of observation (“using the Korean Network for Organ Sharing Database, 2002 to 2016”). This might encourage the reader more.

For example,

The risk factors for early treatment-related mortality in kidney transplantation - A nationwide cohort study

Or The risk factors for treatment-related mortality within first three months after kidney transplantation

Reviewer #2: The paper entitle " The risk factors for treatment-related mortality in kidney transplantation using the Korean Network for Organ Sharing Database, 2002 to 2016", is a well written manuscript, the data is well analysed and proper statistical analysis is applied. All the significant outcomes were well discussed and very comprehensive conclusion was made, I highly recommend this paper for publication.

6. PLOS authors have the option to publish the peer review history of their article (what does this mean?). If published, this will include your full peer review and any attached files.

Reviewer #1: No

Reviewer #2: No

---

## [Author Response · Author response to Decision Letter 0]

29 Oct 2020

We have checked the PLOS ONE style templates and confirmed our manuscript to meet requirements. The file naming also checked.

2. Please include the date(s) on which you accessed the databases or records to obtain the data used in your study.

We accessed the Korean Network for Organ Sharing database on June 5, 2020 to obtain additional information for the revision of a previously reported manuscript associated with treatment-related mortality in kidney transplantation. We found a number of important factors for treatment-related mortality, which could not be integrated in the previous study. Therefore, we have persistently accessed the Korean Network for Organ Sharing database extracted from 40 medical centers to identify the risk factors including routine laboratory data until September 23, 2020. We have added the dates and contact information for a data access committee to the revised Data accessibility statement section (page 20, lines 349 to 353) as follows.

“We had accessed the Korean Network for Organ Sharing database from June 5, 2020 to September 23, 2020. Contact information for a data access committee is listed as follows: National Organ and Blood Management Institute of the Ministry of Health and Welfare, Tel: 82-2-2628-3602; Official internet site: https://www.konos.go.kr/konosis.”

3. Please describe any inclusion and exclusion criteria used to determine your final cohort.

As indicated by the reviewer, we have described inclusion and exclusion criteria, which are similar to our previous study (reference 16) in the revised Materials and Methods section (page 5, lines 98 to 101) as follows.

“This study included all patients enrolled for kidney transplantation in the KONOS system of the KCDC between January 2002 and December 2016. We excluded patients who did not have complete demographic information and who concurrently underwent other organ transplantations [16].”

We have checked the deposited data in Harvard Dataverse. The accessible DOI provided by the Harvard Dataverse was https://doi.org/10.7910/DVN/G4OLYV described in the Data accessibility statement section (page 20, lines 348 to 349) as follows. We have published the dataset immediately and the certificate have been attached to the revised cover letter.

“The data involved in this study have been deposited in Harvard Dataverse, and are accessible through https://doi.org/10.7910/DVN/G4OLYV.”

Response to the reviewer’s comments

1. Is the manuscript technically sound, and do the data support the conclusions?

Reviewer #1: Yes

Reviewer #2: Yes

We have corrected and checked that the revised manuscript described a technically sound piece of scientific research and that the data supported the conclusions. 

2. Has the statistical analysis been performed appropriately and rigorously? 

Reviewer #1: Yes

Reviewer #2: Yes

We have checked that the statistical analysis has been conducted appropriately and rigorously.

3. Have the authors made all data underlying the findings in their manuscript fully available?

Reviewer #1: Yes

Reviewer #2: Yes

According to the journal requirements, we have provided the accessible DOI from the Harvard Dataverse for deposited data. Contact information for a data access committee and the dates obtaining the data were described in the revised Data accessibility statement section. We have added these statement to the revised cover letter.

4. Is the manuscript presented in an intelligible fashion and written in standard English?

Reviewer #1: Yes

Reviewer #2: Yes

We used manuscript editing service before submission.

5. Review Comments to the Author

Response to reviewer #1’s comments

Treatment-related mortality (TRM) after renal transplantation is a concept different from disease-related mortality and appeared to be a very prevalent entity.

Strengths:

- There are scarce data in scientific literature about TRM within 1 or 3 months after kidney transplantation.

- Authors collected an important amount of data from a very large cohort of patients using a national population based database, which included information about a total of 16,073 kidney recipients.

This article would be a valuable contribution to the medical literature to encourage further discussion on this entity.

The writing is clear and easily understandable.

The Authors have worked hard to improve this article, and they have met all criticisms raised by referees.

I feel the manuscript is now suitable for publication in Plos One.

Probably a better title could include the concept of an early mortality after renal transplant, and could exclude the origin of the database and the years of observation (“using the Korean Network for Organ Sharing Database, 2002 to 2016”). This might encourage the reader more.

For example,

The risk factors for early treatment-related mortality in kidney transplantation - A nationwide cohort study

Or The risk factors for treatment-related mortality within first three months after kidney transplantation.

Response 1: As suggested by reviewer, we have changed the title from “The risk factors for treatment-related mortality in kidney transplantation using the Korean Network for Organ Sharing Database, 2002 to 2016” to “The risk factors for treatment-related mortality within first three months after kidney transplantation” in the revised manuscript. 

Response to reviewer #2’s comments

The paper entitle " The risk factors for treatment-related mortality in kidney transplantation using the Korean Network for Organ Sharing Database, 2002 to 2016", is a well written manuscript, the data is well analysed and proper statistical analysis is applied. All the significant outcomes were well discussed and very comprehensive conclusion was made, I highly recommend this paper for publication.

Response 2: Thank you very much for taking the time to review our manuscript. We are very pleased to receive your valuable comment for our manuscript. We hope it will contribute to better outcome of kidney recipients.

We have uploaded our figure files (Fig 1.tif, and Fig 2.tif) to the PACE digital diagnostic tool to meet PLOS requirements. The preview files (Preview_20201028233814619.pdf, and Preview_20201028233901869.pdf) were generated and checked.

---

## [Decision Letter · Decision Letter 1]

24 Nov 2020

The risk factors for treatment-related mortality within first three months after kidney transplantation

PONE-D-20-30070R1

Dear Dr. Jeong,

We’re pleased to inform you that your manuscript has been judged scientifically suitable for publication and will be formally accepted for publication once it meets all outstanding technical requirements.

Kind regards,

Academic Editor

PLOS ONE

Additional Editor Comments (optional):

Reviewers' comments:

Reviewer's Responses to Questions

**Comments to the Author**

1. If the authors have adequately addressed your comments raised in a previous round of review and you feel that this manuscript is now acceptable for publication, you may indicate that here to bypass the “Comments to the Author” section, enter your conflict of interest statement in the “Confidential to Editor” section, and submit your "Accept" recommendation.

Reviewer #1: All comments have been addressed

Reviewer #2: All comments have been addressed

2. Is the manuscript technically sound, and do the data support the conclusions?

Reviewer #1: Yes

Reviewer #2: Yes

3. Has the statistical analysis been performed appropriately and rigorously? 

Reviewer #1: Yes

Reviewer #2: Yes

4. Have the authors made all data underlying the findings in their manuscript fully available?

Reviewer #1: Yes

Reviewer #2: Yes

5. Is the manuscript presented in an intelligible fashion and written in standard English?

Reviewer #1: Yes

Reviewer #2: Yes

6. Review Comments to the Author

Reviewer #1: Treatment-related mortality (TRM) after renal transplantation is a concept different from disease-related mortality and appeared to be a very prevalent entity.

Strengths:

- There are scarce data in scientific literature about TRM within 1 or 3 months after kidney transplantation.

- Authors collected an important amount of data from a very large cohort of patients using a national population based database, which included information about a total of 16,073 kidney recipients.

This article would be a valuable contribution to the medical literature to encourage further discussion on this entity.

The writing is clear and easily understandable.

The Authors have worked hard to improve this article, and they have met all criticisms raised by referees.

I feel the manuscript is now suitable for publication in Plos One.

Reviewer #2: (No Response)

7. PLOS authors have the option to publish the peer review history of their article (what does this mean?). If published, this will include your full peer review and any attached files.

Reviewer #1: No

Reviewer #2: No

---

## [Editor Report · Acceptance letter]

26 Nov 2020

PONE-D-20-30070R1 

The risk factors for treatment-related mortality within first three months after kidney transplantation 

Dear Dr. Jeong:

I'm pleased to inform you that your manuscript has been deemed suitable for publication in PLOS ONE. Congratulations! Your manuscript is now with our production department. 

Kind regards, 

on behalf of

Dr. Robert Jeenchen Chen 

Academic Editor

PLOS ONE